# Projected losses of ecosystem services in the US disproportionately affect non-white and lower-income populations

Jesse D. Gourevitch [1,2✉], Aura M. Alonso-Rodríguez[1,2], Natalia Aristizábal[1,2], Luz A. de Wit[1,2], Eva Kinnebrew[1,2], Caitlin E. Littlefield [2], Maya Moore[1,3], Charles C. Nicholson [1,4,5], Aaron J. Schwartz[1,2] & Taylor H. Ricketts [1,2]

Addressing how ecosystem services (ES) are distributed among groups of people is critical for making conservation and environmental policy-making more equitable. Here, we evaluate the distribution and equity of changes in ES benefits across demographic and socioeconomic groups in the United States (US) between 2020 and 2100. Specifically, we use land cover and population projections to model potential shifts in the supply, demand, and benefits of the following ES: provision of clean air, protection against a vector-borne disease (West Nile virus), and crop pollination. Across the US, changes in ES benefits are unevenly distributed among socioeconomic and demographic groups and among rural and urban communities, but are relatively uniform across geographic regions. In general, non-white, lower-income, and urban populations disproportionately bear the burden of declines in ES benefits. This is largely driven by the conversion of forests and wetlands to cropland and urban land cover in counties where these populations are expected to grow. In these locations, targeted land use policy interventions are required to avoid exacerbating inequalities already present in the US.

[1] Gund Institute for Environment, University of Vermont, Burlington, VT, USA. [2] Rubenstein School of Environment and Natural Resources, University of Vermont, Burlington, VT, USA. [3] Food Systems Program, University of Vermont, Burlington, VT, USA. [4] Department of Entomology and Nematology, University of California, Davis, CA, USA. [5] Department of Biology, Lund University, Lund, Sweden. ✉email: Jesse.Gourevitch@uvm.edu

Nature is essential to human well-being and sustainable development, as decades of research and global assessment have made clear[1,2]. While ecosystem services (ES) have been mapped, modeled, and valued using a multitude of methods and have been studied by a wide range of disciplines, understanding the distribution of ES benefits to individuals and groups within society remains a critical gap[3,4]. As inequality within society becomes one of the most pressing social, political, and environmental issues of this century, this shortcoming has taken on ever more importance for justice and equity in environmental governance and decision-making.

Globally, the projected decline in ES is most severe in developing countries, particularly in Africa and South Asia[5]. Furthermore, local case studies suggest that ES benefits are skewed towards more affluent and less socially vulnerable groups. For example, indigenous people rarely benefit from water quality offset projects associated with road development in the Peruvian Amazon[6]. In the Miyun Reservoir watershed in northern China, poorer households with members who are chronically sick and elderly have less access to ES than wealthier, healthier, and younger households[7]. Even in US cities, the supply of evaporative cooling ES provided by urban vegetation is spatially correlated with neighborhood income, resulting in lower-income communities more exposed to extreme heat[8]. These examples raise concern that natural capital and the ES benefits that flow from it may be just as unevenly distributed as other forms of capital in our society.

Measurement of the distribution of wealth and income has been a prominent area of welfare economics research for decades[9,10], and environmental economics specifically has long focused on the impacts of negative externalities on public goods[11,12]. Moreover, the environmental justice field has documented many instances of minority and lower-income populations being disproportionately exposed to environmental hazards, such as air pollution[13] and natural disasters[14]. However, these types of distributional analyses have rarely been applied to non-market ES benefits[4,15]. To date, ES have most often been quantified in either biophysical terms or in terms of their total economic value[4,16,17]. Both sets of metrics mask the underlying distribution of ES benefits between groups within society and limit our understanding of how ES contribute to human well-being. Without disaggregating ES benefits, it is difficult to identify who wins and who loses in decisions affecting the provision the ES[18]. This information can be used to develop policies that aim to distribute benefits to those most vulnerable to the loss of ES or that facilitate compensation for ES losses.

Multiple conceptual frameworks exist for relating nature's contributions to people[5,19–21], each of which have their advantages and drawbacks. These frameworks often adopt the terms "supply", "demand", and "benefit" in ways that have proven useful for understanding components of ES[16], even if such use does not correspond directly with traditional definitions in microeconomics. We follow this convention and define the "supply" of an ES as a biophysical measure of an ecosystem process or function that has the potential to support or enhance human well-being. We define "demand" for an ES as the need or desire for a good or service, which means demand is therefore predicated on the presence of human populations. Such demand may include mitigation of a potential risk (e.g., negative health outcomes)[22]. Finally, we define the "benefit" of an ES as a function of both supply and demand, whereby a change in human well-being occurs as the result of supply meeting demand. In instances where ES supply occurs in the absence of demand, and vice-versa, there is no benefit.

In the US, as in other parts of the world, projected changes in land cover and population will have major consequences on the supply and demand for multiple ES[23]. As such, the distribution of ES benefits to various groups of beneficiaries could shift dramatically. If historical trends in land cover change continue, projections indicate further loss of natural land cover (i.e., forests, grasslands, and wetlands), with a corresponding expansion of anthropogenic land cover (i.e., croplands and urban areas)[24,25]. In response, the supply of some ES, particularly those not currently valued in markets (e.g., disease risk mitigation), are expected to decline; while others, particularly those valued in markets (e.g., food production), are expected to increase[26,27]. Simultaneously, population in urbanized areas is predicted to grow, while rural populations will shrink[28]. Socioeconomic and demographic groups are also expected to become more segregated on local and regional scales[28,29]. These population shifts have important implications for the magnitude and spatial distribution of ES demand.

Combined, we expect that these changes in land cover and population will create mismatches between ES supply and demand, whereby ES supply decreases in the same locations as where demand increases[16,21,30]. Based on case studies of inequity in the distribution of ES[6–8,31], we hypothesize that such mismatches will disproportionately affect already marginalized groups. To test this hypothesis, we project changes in the supply, demand, and benefits for three ES in every county in the conterminous US between 2020 and 2100. These ES include provision of clean air, protection against West Nile virus (WNV), and crop pollination. We then disaggregate the projected changes in ES benefits across rural and urban communities, socioeconomic (i.e., income quintiles) and demographic groups (i.e., racial groups), and regions of the country (i.e., Midwest, Northeast, South, West). Demographic groups are based on those defined by Hauer[29] and are described in detail in the "Disaggregating Beneficiaries" section in the Methods.

The three ES we model are an illustrative sample of services for which we have data and expertise. In Table 1, we describe who the

**Table 1 Metrics used for estimating ES supply, demand, and benefits.**

| Ecosystem service | Key beneficiaries | Supply metric | Demand metric | Benefit metric |
|---|---|---|---|---|
| Air quality | Downwind population | Avoided PM$_{2.5}$ emissions (kg × yr$^{-1}$) | Downwind population exposed to PM$_{2.5}$ emissions (Count) | Avoided mortalities (Count × yr$^{-1}$) |
| Crop pollination | Farmers within county | Wild-bee abundance (0–1 index) | Pollinator-dependent crop area (ha) | Abundance of wild bees in pollinator-dependent cropland (index) |
| Vector-borne disease control | County population | Avoided risk of exposure to West Nile virus (Count per 100,000 people × yr$^{-1}$) | Population exposed to West Nile virus (Count) | Avoided cases of West Nile virus (Count × yr$^{-1}$) |

See Methods section for more information about the data and models used to estimate each of these metrics. Social inequalities may be reflected in how ecosystem services are distributed among groups of people. Here the authors estimate the distribution of three ecosystem services across demographic and socioeconomic groups in the US between 2020 and 2100, finding that non-white and lower-income groups disproportionately bear the loss of ecosystem service benefits.

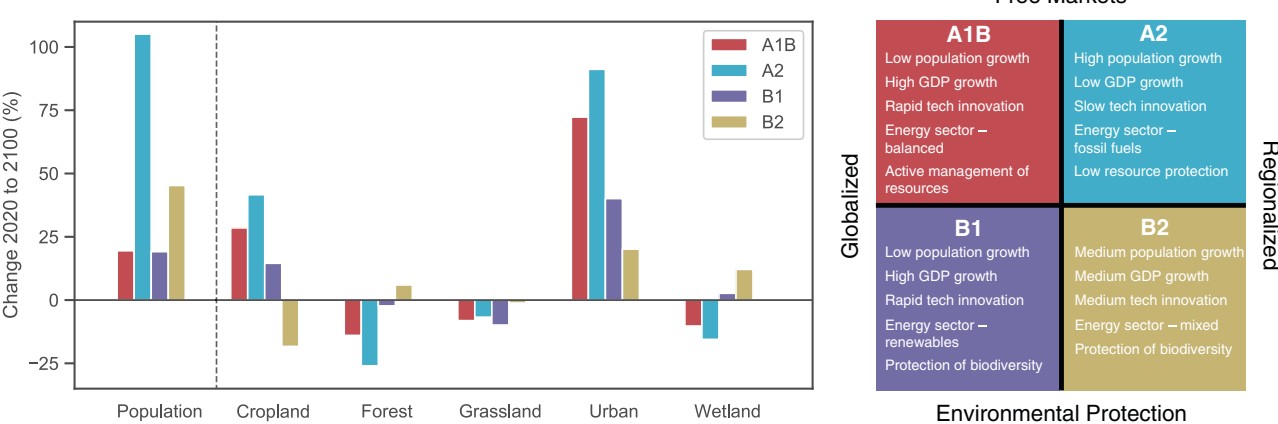

**Fig. 1 Projected changes in national population and land cover between 2020 and 2100.** Projections vary by IPCC SRES scenario, as indicated by the color of the bars. See EPA ICLUS (2008) for more information about population projections and Sohl et al. (2012) for more information about land cover projections.

beneficiaries are for each ES and the metrics used for quantifying supply, demand, and benefit. The beneficiaries, the spatial scale over which the benefits are realized, and the way benefits are accrued vary across ES. For air quality and protection against WNV, the beneficiaries include all households within a spatial unit. Changes in the supply of protection against WNV only affect the population within a given county, whereas changes in air quality have regional impacts downwind of where emissions occur.

Our ES models are driven by existing datasets predicting future changes in land cover and population across the US between 2020 and 2100. While several land cover and population projections exist[29,32], we select datasets with common underlying assumptions and that are used by US federal government agencies. For land cover projections, we use the United States Geological Survey (USGS) FOREcasting SCEnarios of Land-Use Change (FORE-SCE) dataset[33,34] (Fig. 1). For population projections, we used the United States Environmental Protection Agency (US EPA) Integrated Climate and Land Use Scenarios (ICLUS) dataset[35] (Fig. 1). The population projections are then coupled with county-level income[28] and demographic[29,36] projections. These projections are modulated by four alternative future scenarios (Fig. 1), specified by the IPCC Special Report on Emissions Scenarios (SRES). See Methods for more details on the assumptions and development of these scenarios.

This study projects changes in multiple ES benefits at the national scale and assesses the distribution of those benefits among demographic and socioeconomic groups. Calls have been made for better disaggregation of ES benefits among beneficiary groups[3,37,38], yet to date, relatively few studies have done so[4]. Our analysis builds on prior studies by disaggregating ES benefits according to specific racial groups, as well as income quantiles. This is particularly important in the US, where historical political, economic, and social trends have perpetuated and reinforced inequality specifically along lines of race and class. In general, we find that declines in ES benefits between 2020 and 2100 disproportionately affect non-white and lower-income populations. These trends raise concern that the projected shifts in land cover and population may exacerbate existing inequalities in the US.

## Results
**Aggregate trends.** As a result of land cover change and population shifts between 2020 and 2100, the US will experience declines in ES benefits under nearly all scenarios (Fig. 2). The magnitude of these trends varies by ES and by scenario. In general, declines

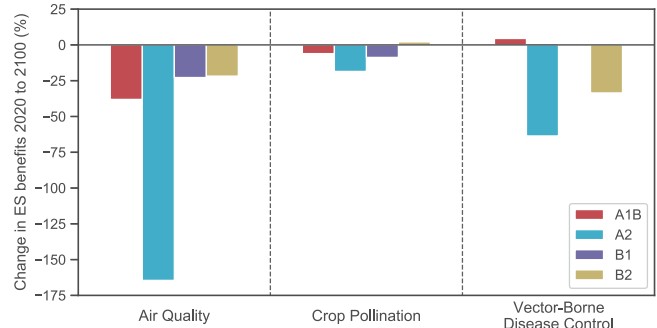

**Fig. 2 Projected changes in ES benefits between 2020 and 2100.** Benefits for each service were calculated at the county-level, then summed at the national level. Projections vary by IPCC SRES scenario, as indicated by the color of the bars. The indicators used for each of the benefits are described in Table 1.

in ES benefits are greatest under scenario A2 and are mitigated under scenarios B1 and B2. In scenario A1B, WNV disease control increases by 4.5%. Hereafter, we focus our results on scenario A2, as observed $CO_2$ emissions between 1990 and 2010 are closely aligned with projections made under this scenario[40]. Results for scenarios A1B, B1, and B2 are shown in the Supplemental Information (Supplemental Fig. 1).

**County-level trends.** At the county-level, expected changes in ES supply and demand are highly variable (Fig. 3; Supplementary Fig. 1). ES mismatches are expected to occur in counties where supply decreases and demand increases between 2020 and 2100 (i.e., purple-colored counties, Fig. 3; Supplementary Fig. 1). Among these counties, the severity of ES mismatches depends on the relative magnitude of changes in ES supply and demand.

Although regional and state-level trends are less apparent, county-level changes in supply and demand illuminate some of the drivers underlying the aggregate national statistics. For example, both supply and demand for air quality decrease in most counties (Fig. 3A, orange-colored counties) as cropland and urban land cover increases (i.e., land uses associated with greater emissions) and rural counties depopulate. However, demand for improved air quality increases in urban counties, due to projected growth in population density; the outsized decline in air quality benefits in these counties results in a net loss of benefits nationally.

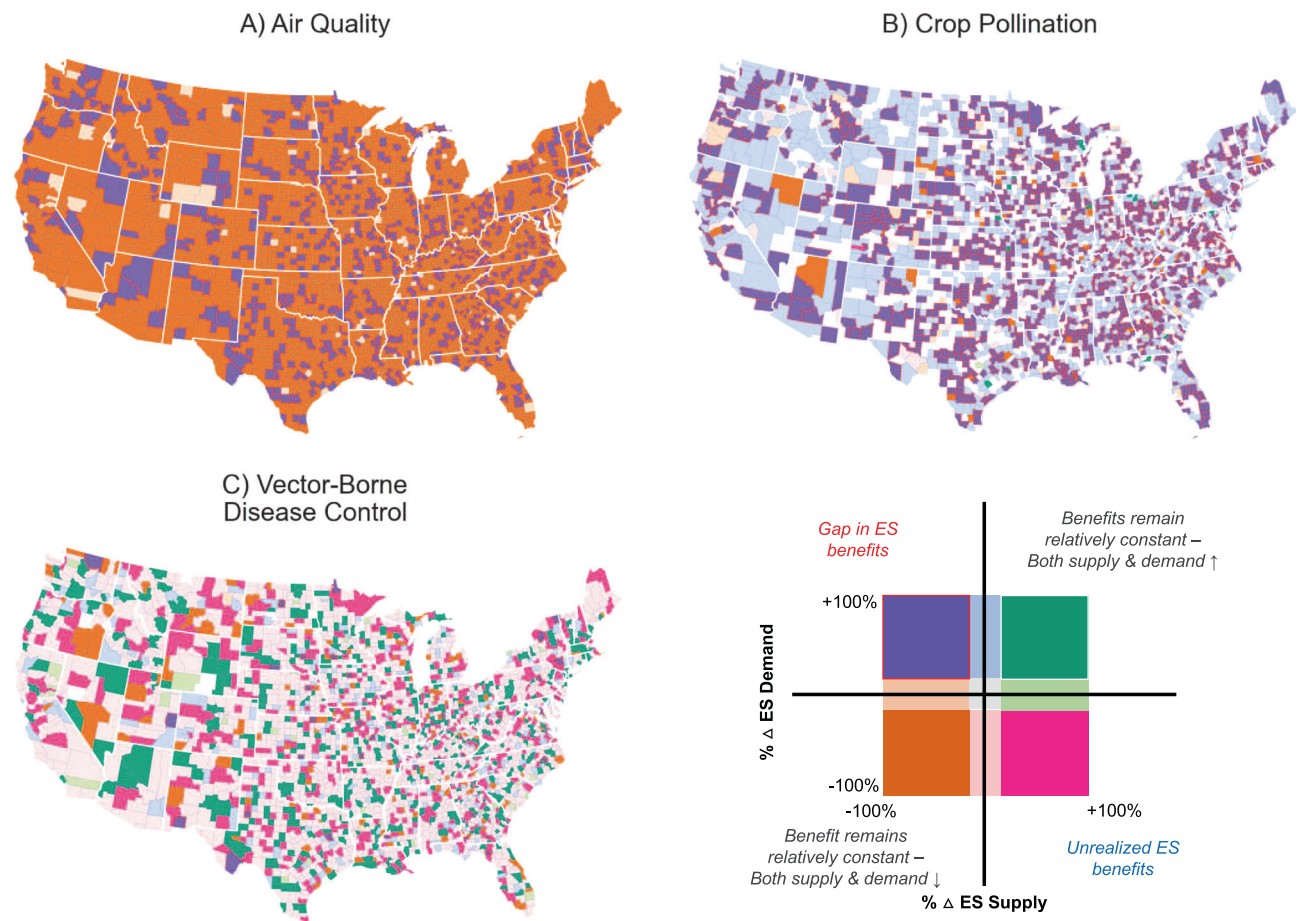

**Fig. 3 Maps of changes in ES supply and demand between 2020 and 2100 at the county-level for scenario A2.** Counties where supply or demand change between −5 and 5% are plotted with lower hues. Purple areas outlined in red indicate counties where supply and demand mismatches are expected to occur. This color scheme does not distinguish the severity of projected mismatches between ES supply and demand. For instance, a 90% decrease in supply and a 90% increase in demand will result in a greater mismatch than a 10% decrease in supply and a 10% increase in demand.

For crop pollination, demand among farmers increases in most counties, while supply decreases (Fig. 3C, purple-colored counties). These changes vary spatially, but in scenario A2, most counties are expected to experience an increase in pollinator-dependent cropland. As the proportion of crop types grown in each county are assumed to remain constant, this increase in demand occurs due to expansion of cropland area (see methods for more details). This expansion, in part, also coincides with a loss of forested land area, which contributes to a decline in crop pollination supply.

Similar to air quality, demand for vector-borne disease control decreases in a large majority of counties, but increases in counties containing urban centers (Fig. 3D). Supply of vector-borne disease control in most counties either remains relatively constant or increases, leading to unrealized ES benefits. Due to differences in habitat preference for WNV vectors (i.e., mosquitos), the land cover conversions driving the increased supply of vector-borne disease control vary by region (Supplementary Fig. 4). For example, increased forest cover in the Great Plains is associated with lower risk of WNV, but in the Eastern Temperate Forests, greater forest cover is associated with elevated risk of WNV.

**Distribution among groups.** We find stark differences in the distribution of changes in ES benefits among beneficiary groupings (Fig. 4). Across all services the directionality of change is opposite for rural and urban counties (Fig. 4; top row). In rural

counties, benefits of air quality and vector-borne disease control increase between 2020 and 2100, while crop pollination benefits decrease. Those trends are reversed for urban counties. Compared with rural and urban counties, suburban counties are predicted to experience relatively little change.

Among income groups, counties in the lowest quintile are projected to experience the greatest losses in air quality and WNV benefits (Fig. 4; second row). By contrast, counties in the highest quintile are predicted to gain benefits for air quality and vector-borne disease control, but farmers in those counties are expected to experience declines in crop pollination. For counties in the 2nd and 4th quintiles, the magnitude of changes is smaller than in the 1st and 5th quintiles. Counties in the 3rd quintile experience relatively little change compared to other counties.

Similar to the trends across income groups, the changes in ES benefits for non-white groups are the opposite of the trends for white communities (Fig. 4; third row). In particular, Black and Hispanic people are expected to experience substantial losses in ES benefits, while white people will experience moderate gains. Averaged across scenarios, air quality, crop pollination, and vector-borne disease control will decrease for non-white people by 224%, 118%, and 111%, respectively. For white people, the benefits from these ES will increase by 10%, 35%, and 36%.

Differences in how ES benefits are distributed across broad geographic regions of the US are less apparent than the other beneficiary groupings (Fig. 4; bottom row). For air quality, there is a consistent decline in benefits in each of the four broad regions

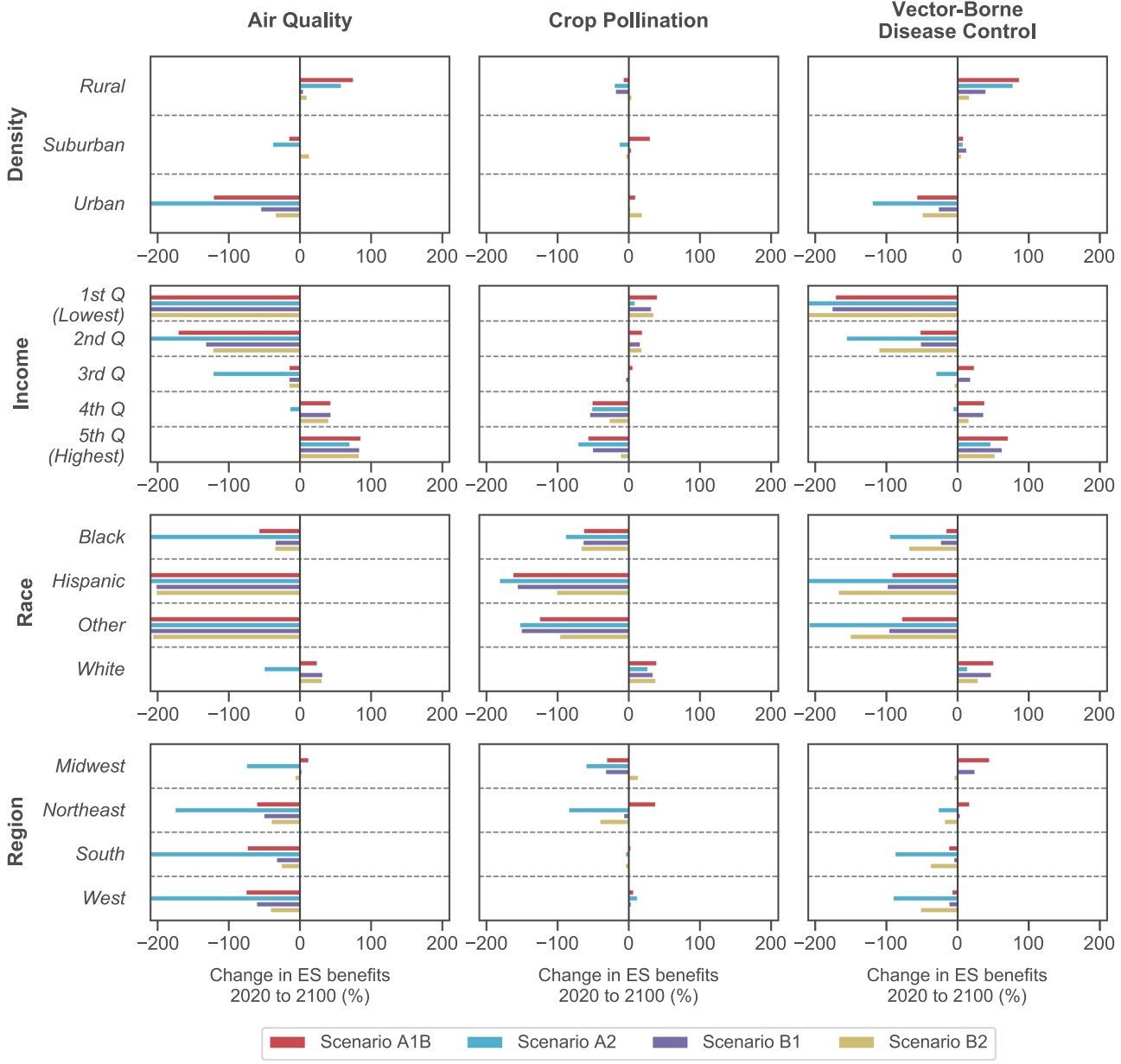

**Fig. 4 Distribution of percent changes in ES benefits, between 2020 and 2100, across various groups of beneficiaries.** Projections vary by IPCC SRES scenario, as indicated by the color of the bars. The lower and upper limits of the x-axis for each subplot are set to −210 and 210 to facilitate comparison across subplots; however, some bars extend beyond these bounds.

of the US (i.e., Midwest, Northeast, South, West). However, vector-borne disease control benefits increase in the Midwest and Northeast under scenarios A1B and B1, while benefits decrease in the South and West. For crop pollination, the magnitude of change is relatively large for farmers in the Midwest and Northeast.

## Discussion

These results show that ES benefits in the US will not only decline between 2020 and 2100, but also that those declines will most severely affect already marginalized communities. Other studies have also found that declines in ES will be likely at national and global scales[5,26]; however, in this study, we disaggregate results within a country to show that non-white, lower-income, and urban communities are at the greatest risk of losing benefits from ES. These findings complement widespread evidence that

marginalized people are disproportionately exposed to environmental hazards[41]. Our findings raise concerns that, if left unattended, these environmental inequalities will persist and worsen over the next century.

In most ES studies to date, ES benefits are either disaggregated spatially (e.g., pixels across a landscape), regionally, or not at all, and are rarely disaggregated among beneficiary groups[4]. Although spatial segregation along race and socioeconomic lines is a defining feature of American landscapes, these divides typically occur within smaller spatial scales, such as cities or counties. Here we show that the differences in projected ES benefits among regions of the country are relatively small (Fig. 4), and that even the results mapped at the county-level show weak spatial patterns (Fig. 3). Instead, the largest differences in projected ES benefits occur between income and racial groups. When beneficiaries are grouped by administrative or even watershed boundaries, as previous ES studies have done, inequitable distributions of ES

benefits may be masked. In recognition of this shortcoming, several recent studies have demonstrated the potential for novel tools and techniques, such as open-access social media datasets (e.g., Hamstead et al.[42]), agent-based modeling (e.g., Miyasaka et al.[43]), and social vulnerability indices (e.g., Mullin et al.[39]), in evaluating the distribution of ES benefits among beneficiary groups[38].

The disproportionate impact of ES losses on marginalized people is largely due to the conversion of natural land cover in counties where these communities are expected to grow between 2020 and 2100. With the exception of WNV, ES supply declines in counties where forests and wetlands are converted to cropland and urban land cover (Supplementary Fig. 3). Forests provide nesting habitat for insect pollinators, while fertilizer and tailpipe emissions from cropland and urban areas threaten air quality. By contrast, the relationships between risk of WNV and change in land cover are not generalizable on a national-scale because habitat preference for vectors of WNV (i.e., mosquitos) varies by ecoregion (see Supplementary Fig. 4)[44]. As the US becomes increasingly less white and income groups become more geographically segregated, the decline in the supply of ES are expected to largely impact non-white and low-income communities.

In the loss or absence of ES benefits, a range of outcomes or responses may occur. The most obvious consequence is a direct cost or damage to a community. For example, if air quality benefits decrease, then premature mortalities caused by respiratory diseases may increase, or if crop pollination decreases, then yields for pollinator-dependent crops may decline. Alternatively, communities may respond by substituting the ES with a non-nature-based solution. For instance, if vector-borne disease control benefits decrease, then communities may respond by spraying pesticides in order to reduce incidence of WNV. However, the cost-effectiveness, efficiency, and accessibility of ES substitutes remains a critical area of future research[45].

Emigration is another possible response to declines in the supply of ES[46]. For instance, variation in local and regional air quality has been found to induce household migration[47]. Similarly, increasing frequency and severity of flooding has triggered voluntary buyouts of flood-prone properties[48]. These examples raise concern that the decrease in ES supply in one locale may simply displace the demand for that ES from that location to another. In the case of government-sponsored buyout programs, decisions regarding which properties to acquire have major social justice implications[49]. However, when migration occurs in the absence of government assistance, the financial costs to households are often overwhelming, resulting in further vulnerability to environmental hazards and the loss of ES.

Throughout our analysis, we evaluate changes in ES benefits relative to a reference state representing conditions in 2020. However, our results do not imply that the US is moving from an equitable baseline to an inequitable future. Instead, such inequities are assumed to be present and, in part, likely underpin the disparities projected here. The current distribution of ES benefits must therefore be understood within the context of historical, political, and economic forces that have reinforced environmental privilege and perpetuated class and race-based injustice. For instance, Black and Hispanic people are currently disproportionately exposed to air pollution caused mainly by the consumption of goods and services by white people[50]. That disparity, and others, are only expected to worsen (Fig. 4).

Crop pollination presents an interesting contrast with the other two ES. Unlike the other ES, crop pollination services are mediated by agricultural markets, whereby farmers (as opposed to all households) within a given county are the most direct beneficiaries. While consumers of pollinator-dependent crops also benefit from pollination services, there is of course a major disconnect between the locations where crops are produced and where they are consumed. Addressing this complexity is beyond the scope of our analysis.

Our findings contain several important areas of uncertainty. First, the land cover and population projections used are not intended as best-estimates of future trends. Instead, they represent a range of possible outcomes based on various assumptions regarding economic development, material consumption, fertility rates, population movement, and environmental governance. Further, the lack of consensus among other land cover projections raise concerns about the validity of these modeling approaches generally[32]. As compared to other land cover projection datasets based on the SRES scenarios, such as those developed by Wear[51], Bierwagen, et al.[35], and Strengers, et al.[52], the FORE-SCE model projections[33,34] we use encompass greater variability across scenarios. We therefore would assume that our results include a wider range of possible outcomes than if we were to use of these other datasets. Second, projecting ES at the national scale required the use of simple modeling approaches, broadly available datasets, and generalized parameters. As is true in any modeling application, there is a trade-off between model scalability and complexity. Our approaches tend toward the scalable, in order to address these issues across the entire conterminous US.

Validation, calibration, and uncertainty assessment of ES models is a critical area for further research. Better understanding the uncertainties in model-based predictions of ES supply and demand has the potential to increase the credibility of this information, and increase its relevance in decision-making[53]. Although there are certainly unique challenges in validating and calibrating ES models, there are also many opportunities to apply existing methods from other fields to ES assessments[54]. Empirical monitoring of changes in ES benefits is another important aspect of ES model validation. This could be done directly by measuring changes in human well-being, such as incidence of WNV, or indirectly by measuring the value of ES substitutes, such as the price of honeybee hives.

Capturing the impacts of climate change on land cover change, ES supply, and ES demand is another important area of future research. Neither of the land cover or population datasets we use explicitly account for climate change in their projections, and in none of our ES models do we include climatic variables. Despite these limitations, increasing evidence suggests that climate change adaptation may lead to significant changes in land use, with corresponding impacts on ES[55,56]. Moreover, climate change is expected to negatively affect ecological processes mediating ES supply[57,58], as well as shift ES demand through migratory responses of beneficiaries. Separate from the direct impacts on ES, damages from climate change are expected to disproportionately affect poorer regions of the US, thus increasing preexisting inequality[59]. Combined, these trends raise concern that our results indicating future disparities in ES are likely underestimates.

Our study provides evidence of social-environmental challenges for this century, but it also presents opportunities. It is a call for land owners, researchers, and decision-makers to implement policies and practices that prevent further inequitable distribution of ES benefits. Specifically, our results show that there is an opportunity for conservation organizations and urban planners to better integrate equity and justice into every facet of their work, and for social justice groups to consider the role of conservation and land-use policy in reducing inequality. Spatial targeting of conservation interventions will not only need to consider where the supply of ES is threatened, but also where demand for ES is greatest and by which groups of people. For instance, promoting agroecological practices that provide habitat for pollinators are needed not only where crop pollination gaps

are highest, but also for farmers who lack access to substitutes for wild-bee pollination. Similarly, federal payment for ecosystem programs, such as the Conservation Reserve Program, could target payments not only in locations where the supply of ES is most at risk, but also where people are most vulnerable to the loss of ES. Together, these types of changes in land-use policy and practices have the potential to protect, restore, and redistribute ES benefits where they are most needed.

## Methods

**Future scenarios**. Both land cover and population datasets are modulated by alternative socioeconomic and climate scenarios, representing various pathways through which social, economic, political, and environmental trends may affect land cover and population trajectories. These scenarios make varying assumptions about the degree to which the economy is globalized and regulated for environmental protection, and have implications for population growth rates, GDP growth rates, technological innovation, energy sources, and natural resource protection (Fig. 1). The FORE-SCE and ICLUS datasets are based on the IPCC Special Report on Emissions Scenarios (SRES), while the income and demographic projections are based on the Shared Socioeconomic Pathways (SSPs). To ensure consistency across datasets, we map the SSPs onto the SRES based on recommendations presented by van Vuuren and Carter[60].

The SRES include four alternative scenarios: A1B, A2, B1, and B2. Scenario A1B assumes rapid economic expansion, relatively limited population growth, and development of more efficient technologies; scenario A2 assumes regionalization of economic activity, high population growth, and extensive fossil fuel use; scenario B1 makes similar assumptions as A1B except that economic activity shifts towards service and information-based industries and greater emphasis is placed on environmental sustainability; scenario B2 is similar to A2 except that material consumption and fossil fuel extraction declines[61]. Under scenarios A1B, A2, and B1, cropland and urban land-use area increase at the expense of forests, grasslands, and wetlands, with A2 projecting the most extreme shifts (Fig. 1). By contrast, Scenario B2 predicts relatively minor increases in urban land cover, while declines in cropland area are offset by gains in forests and wetlands.

**Disaggregating beneficiaries**. For each service, we disaggregated projected changes in benefits across rural and urban communities, socioeconomic groups, regions of the country, and demographic groups. For all categories, except demographic groups, counties are grouped into discrete bins. Counties with less than 10,000 people are considered rural, counties with populations between 10,000 and 50,000 are considered suburban, and counties with more than 50,000 people are considered urban[62]. Socioeconomic groups are based on the household-level income, whereby each county is assigned to a quintile based on its per capita income. Regional classifications are based on state (Supplementary Table 1).

Projections of demographic change are drawn from the dataset developed by Hauer (2019)[29]. Following the classifications used in this dataset, we define demographic groups based on the population of each county that is Black (non-Hispanic), Hispanic, Other (non-Hispanic), and White (non-Hispanic). Hauer[29] states that the Other (non-Hispanic) group refers specifically to "American Indian/Alaska Native and Asian/Pacific Islander" populations. Moreover, each group is mutually exclusive of the others, such that there is no overlap between groups. In contrast to the other categories of disaggregation, we partition ES benefits for a county based on the relative proportion of each demographic group within that county. For example, if cases of WNV in a county are expected to decrease by 10 and that county is comprised of 70% Black people and 30% white people, then we estimate that 7 fewer Black people and 3 fewer white people will become infected with WNV. This approach does not account for heterogeneity of impacts within counties.

We acknowledge here that grouping people based on race is imperfect, as there is no biological basis for defining differences by race. In our results (see Fig. 4), however, there are distinct trends differentiating predicted outcomes for the White group, versus outcomes for the Black, Hispanic, and Other groups. For summary purposes, we refer to Black, Hispanic, and Other people under a single umbrella term: "non-white". While the term "non-white" may not be appropriate in many circumstances, we find it to be useful here specifically because the predicted outcomes for the White group are directionally opposite from the outcomes for the Black, Hispanic, and Other groups.

To account for racial disparities between farmers and the general population of the counties in which they operate[63], we adjusted the demographic projections as they relate to the beneficiaries of crop pollination. By comparing the percentage of white farm operators in each county (USDA NASS Census 2012) with percentage of white people in the total population of the county (US Census 2012), we found that farm operators are on average 15% more white (Supplementary Fig. 2). Given this disparity, we applied county-specific scalars to adjust the population demographic populations accordingly. For example, take a county where the population is 50% white, 30% Black, and 20% Hispanic; when recalculated, those proportions change, such that the farmer operator demographics are 57.5% white, 25.5% Black, and 17% Hispanic. As a result of this adjustment, white populations are expected to be more greatly affected by changes in crop pollination benefits. Farmer income is also on

average lower than the general population; however, because we bin counties by income quintile, we assume that the spatial distribution of farmer incomes matches that of the general population (e.g., low-income farmers operate in lower-income counties, and vice-versa).

**Air quality**. We applied the Intervention Model for Air Pollution (InMAP) to estimate changes in air quality under each land cover scenario. InMAP (http://spatialmodel.com/inmap) is an open-source, spatially explicit chemical transport model that simulates the annual average transport, transformation, and deposition of air emissions[64]. InMAP is more computationally efficient than other chemical transport models and only requires the input of the total annual emissions at a source location. In our analysis, source locations are represented as counties.

Potential changes in $NO_x$, $NH_3$, $SO_2$, and VOC emissions in each county were estimated by multiplying the area of each land cover class within the county by associated emissions factors drawn from the Model of Emissions of Gases and Aerosols from Nature (MEGAN) v2.1[65] and the US National Emissions Inventory[66]. Emission factors for each land cover class are shown in Supplementary Table 2. Forests, grasslands, and cropland naturally emit VOCs; $NH_3$ emissions result from volatilization of ammonium-based agricultural fertilizers in cropland; $NO_x$ is also emitted from denitrification of nitrogen-based fertilizers and is produced by combustion reactions (i.e., automobile engines); $SO_2$ emission results from combustion of fossil fuels. This approach assumes that all locations of the same land cover type have equal emissions.

InMAP assumes linear relationships between $NO_x$, $NH_3$, VOC, and $SO_2$ emissions and ground-level deposition of $PM_{2.5}$ and $O_3$. We estimated the marginal damages of emissions by running InMAP for each county based on an assumed change of one unit of $NO_x$, $NH_3$, VOC, and $SO_2$ emissions. The model outputs a receptor matrix shapefile covering the entire US, where the size of the receptor cells varies depending on population density. Within each receptor cell, InMAP estimates elevated PM2.5 and $O_3$ concentrations. We aggregated receptor cells to the county-level.

To predict damages to human health from $NO_x$, $NH_3$, VOC, and $SO_2$ emissions loss attributable to changes in land cover within each county, we coupled InMAP outputs for elevated concentrations of PM2.5 and $O_3$ and the estimated population within each county. Changes in the relative risk of premature mortality from all causes of mortality due to exposure to elevated PM2.5 concentrations were estimated using the Cox proportional hazards function, as described by Lin and Wei[67]. This model assumes that the relative risk of premature mortality increases exponentially with increases in $PM_{2.5}$ concentrations. We calculated the increase in mortality due to elevated $PM_{2.5}$ according to the methods described by Tessum, et al.[68], where the expected increase in premature mortality is multiplied by the affected population size and the baseline rate of mortality.

**Crop pollination**. We focus on bees as pollinators, given their role in pollinating more than 80% of all flowering plant species[69] and as key ES providers for the majority of crop species worldwide[70]. We used the Lonsdorf et al. model (LEM) to calculate the supply of pollination services by bees. This spatially explicit model of wild-bee visitation has been described by Lonsdorf, et al.[71] and validated with field visitation observations[72,73]. In brief, the LEM maps relative abundance of nesting pollinators, then models visitation as the distance-weighted average abundance of surrounding nests. With increasing distance from nest sites in all directions, the model assumes an exponential decay in visitation. The model produces a relative index (0–1) of pollinator visitation.

The LEM requires gridded rasters of floral and nesting values. For this input, we associated each land cover class with expert-opinion derived floral and nesting values[74]. The model's single parameter, α, is a distance decay scalar representing the average distance a bee would travel to forage. Based on a previous meta-analysis[75], we used an average foraging distance of 600 m for temperate wild bees. We applied the LEM to each land cover scenario, then summarized pollination supply as the relative visitation of wild bees for each US county by averaging the visitation index for all cropland pixels within that county.

We calculated the demand for pollination services for each US county as the area of pollinator-dependent crops. We used the 2017 National Agricultural Statistics Service census[76] to determine the area for all disclosed crops within each county. We used the median pollination dependency value in the range reported by Klein, et al.[70] for each crop type. We then calculated demand-weighted crop area for each county as: $D_{co} = \sum_{i=1}^{n} D_i A_i$ where $D_i$ is the pollinator dependency rate of crop i and $A_i$ is the area of crop i. To estimate pollination demand in future years, we used this value to determine the dependency-weighted proportion of crop area per county in 2020, and multiplied this proportion by the cropland area in each future year. The proportion of pollinator-dependent crop area is therefore assumed to remain constant over time, but as cropland area fluctuates across years, the absolute area of pollinator-dependent crop area varies. While demand for specific crops is likely to change in future years, modeling how the proportion of crop types are expected to change through time is beyond the scope of our analysis. To estimate the benefit of pollination services, we multiplied the supply and demand metrics.

**Vector-borne disease control**. We modeled vector-borne disease control as it related to incidence of West Nile Neuroinvasive Disease (WNV). To predict changes in WNV, we developed statistical models incorporating proportions of

each land cover class within each county, controlling human population density. We trained the models using the observed annual average WNV cases (per 100,000 people) by US county between 2006 and 2016[77] and average proportions of land cover types, based on the National Land Cover Dataset for the same years.

Given the variability in WNV host and pathogen habitat across the US, we divided the conterminous US into Ecoregions of North America Level 1 as defined by the US Environmental Protection Agency[78], assigning each county to the Ecoregion classification that represents most of its area, and further built a predictive model for WNV incidence for each ecoregion. We also included human population density in the model to control for its effect on the probability of disease occurrence. We calculated human population density by county by dividing total county population by county area. This approach does not capture other factors affecting risk of WNV, such as climate, socioeconomic factors, and more granular habitat features (e.g., abundance of standing freshwater)[79,80].

For each ecoregion we excluded land cover types with correlation coefficients greater than 0.5 and sequentially prioritized land cover types as follows: Forest, Urban, Cropland, Wetland, Grassland, Water, Shrubland, and Barren, based on area extent, human risk of exposure and potential WNV habitat. For all ecoregions, human population density was highly correlated with Urban land cover type (r ≥ 0.5) and was thus excluded from all models. We used the glmulti package in R to select the best fitting model for each Ecoregion based on the lowest Akaike information criterion (AIC) value. We then fitted WNV incidence using generalized linear models with Gamma distributions. We validated each model using the cv.glm function in the boot package in R. Predictive accuracy varied by model, averaging 62% and ranging from 33% in the Great Plains Ecoregion to 97% in the Northern Forest Ecoregion.

We projected WNV incidence, also referred to as "realized incidence" in response to land-use change for the years 2020 through 2100 using the fitted models with only the land cover predictor variables that were statistically significant at an alpha level of <0.1. We considered avoided risk of exposure to WNV (Count per 100,000 people × yr$^{-1}$) to represent supply of reduced risk of exposure (i.e., not accounting for population exposure to the virus), while the human population projected by each scenario represented the demand. By multiplying the supply and the demand for each county and dividing by 100,000, we calculated the benefit as the inverse of the realized number of WNV cases.

**Reporting summary**. Further information on research design is available in the Nature Research Reporting Summary linked to this article.

## Data availability
The output data that support the findings of this study are available in a FigShare repository with the following identifier: https://doi.org/10.6084/m9.figshare.13622774.

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

## Acknowledgements

We would like to thank Rebecca Chaplin-Kramer, Laura Sonter, and Brendan Fisher for providing thoughtful comments and feedback on earlier versions of the manuscript. We would also like to thank the Gund Institute for Environment at the University of Vermont for providing institutional support. J.D.G. is supported by the National Science Foundation under the Vermont EPSCoR program [grant numbers EPS-1101317 and NSF OIA 1556770]. Any opinions, findings, conclusions, or recommendations expressed in this material are those of the authors and do not necessarily reflect the views of the National Science Foundation or the Vermont EPSCoR program.

## Author contributions

J.D.G.: Conceptualization, Methodology, Software, Formal Analysis, Investigation, Data Curation, Writing—Original Draft, Writing—Review and Editing, Visualization, Supervision, Project Administration; A.M.A.-R.: Conceptualization, Methodology, Formal Analysis, Investigation, Writing—Review and Editing; N.A.: Conceptualization, Methodology, Investigation, Investigation, Writing—Review and Editing; L.A.d.W.: Conceptualization, Methodology, Software, Validation, Formal Analysis, Investigation, Writing—Original Draft, Investigation, Writing—Review and Editing; E.K.: Conceptualization, Methodology, Investigation, Writing—Original Draft, Investigation, Writing—Review and Editing; C.E.L.: Conceptualization, Methodology, Investigation, Writing—Original Draft, Investigation, Writing—Review and Editing; M.M.: Conceptualization, Methodology, Investigation, Investigation, Writing—Review and Editing; C.C.N.: Conceptualization, Methodology, Software, Formal Analysis, Investigation, Writing—Original Draft, Investigation, Writing—Review and Editing; A.J.S.: Conceptualization, Methodology, Investigation, Investigation, Writing—Review and Editing; T.H.R.: Conceptualization, Resources, Investigation, Writing—Review and Editing, Supervision, Project Administration.

## Competing interests

The authors declare no competing interests.
