## [Peer Review File · Nature Communications]

Reviewer comments, first round -

Reviewer #1 (Remarks to the Author):

This paper examines how projected changes in U.S. ecosystem services affect different demographic groups during the period 2020 to 2100. The authors combine government projections of land-use with similar projections of population and income, and then translate the land-use change projections to a set of ecosystem services – carbon storage, air quality, crop pollination, and West-Nile virus protection. Results indicate that projected losses in ecosystem service will disproportionately affect non-white, low-income groups, especially in rural regions. This large-scale analysis of the effects of ecosystem services on different demographic groups is novel in the literature, and sheds light on an issue that is important to conservation science. I am generally positive about the manuscript, though there are multiple issues that need to be addressed. Main comment #1: carbon storage and crop pollination services should be dropped from the analysis, which should be re-focused on the two ES that have distributional consequences and which best suits the authors' methods.

- Crop pollination services

- o The beneficiaries are farmers. The Census of Agriculture (2017) says 95.4% of U.S. producers are white (https://www.nass.usda.gov/Publications/Highlights/2019/2017Census_Farm_Producers.pdf). But the U.S. Census says only 76% are white.

- o The authors appear to use the population demographics to apportion the projected changes in crop pollination services (Fig. 4), but this is quite wrong since most farmers are white and it's unclear how the demographics of farmers differs from the counties where they reside.

- o A similar problem occurs with income, where it is not clear that income of farmers matches that of the population, and yet that's what appears to be assumed in Fig. 4. The USDA Economic Research Service has more information on incomes of farming households that might be useful (<https://www.ers.usda.gov/data-products/ag-and-food-statistics-charting-the-essentials/farming-and-farm-income/>).

- o Given how the authors are computing the distributional aspects of changes in ES, and given that the beneficiaries of crop pollination services are farmers whose demographics differ from the broader population, I think the authors should consider dropping crop pollination services unless they can develop a more accurate way to measure demographics of this sub-group.

- Carbon storage

- o I don't see why the authors include this service, since the beneficiaries include the global population. The portion of Fig. 3 for carbon storage simply presents the areas where carbon storage is projected to decline. But this has already been analyzed in past papers (e.g. the Lawler et al. 2014 study you cite), and the present manuscript doesn't really add anything new.

- o My suggestion is for the authors to drop carbon storage since it's not a part of the distributional effects of ES change (Fig. 4), which is this paper's main story.

Main comment #2: The land-use and demographic projections that the authors use have major limitations that could be better discussed. I know that analyses of this scale must hold their nose and make huge assumptions (I've done the same), but I do think more discussion could be added to give the reader a better intuitive sense of the importance of these assumptions.

- Migration response to changes in ecosystem services

- o This needs more enhanced discussion, as it something ignored in your modeling.

- o The economics literature has produced much evidence that people's location decisions are related to local amenities, and air quality has received particular attention (Sieg et al. 2004; Bayer et al. 2009). Reductions in air quality in location X relative to location Y induce migration from X to Y, and corresponding housing price reductions in X and increases in Y.

- o Your current framework ignores such location adjustments. I understand why you ignore such adjustments (which would be really challenging at this scale), but I think this needs a more extended discussion. Using your language, a reduction in the "supply" of an ES in a region might induce a reduction in the "demand" of an ES by inducing out-migration or lower in-migration to the same region. How this would play out in terms of distributional effects would be a fruitful future

research topic – would it exacerbate the distributional effects? E.g. past literature indicates that pollution increases induce out-migration and corresponding changes in housing prices, which will feed back and influence population and demographic projections.

- Choice of land-use change projection and no link to climate change
 - o The Sohl et al. (2016 Ecol Modeling) paper you cite notes that land-use projections vary significantly across different models and methods. I suggest adding a discussion of how the FOR-SCE projections might give different answers compared to some other candidates discussed in that article. Based on the Sohl et al. article, can you say whether other land-use projections would give more optimistic or pessimistic results in terms of disproportionate impacts? This deserves more mention than your one-sentence on p. 6 about “uncertainty”.
 - o I’m glad that you note that the lack of climate change impacts on the land use projections is a problem. There is, as yet, no large-scale land-use projections linked to climate change for the U.S. However, you should note that there is progress on this in the UK (Fezzi et al. 2015), and there is work that models land-use change within western U.S. forests in response to climate change adaptation by landowners (Hashida and Lewis 2019), with corresponding impacts on ecosystem services (Hashida et al. 2020).. For the U.S., there is also evidence that climate change will alter urban quality of life (Albouy et al. 2016), with potential for shifting migration patterns related to my earlier comment about migration. And there has been a very active economics literature on climate impacts on farmland values (e.g. see the review by Mendelsohn and Massetti 2017; or recent paper by Ortiz-Bobea 2020). So, we know that economic returns to different land uses affect land-use change (Lawler et al. 2014), and the above literature indicates that climate change could affect land use decisions by altering these returns in a variety of directions that the FOR-SCE projections ignore. An intuitive mention of the land-use climate change link could really help the reader understand the limitations of your selection of land-use projections.

References cited in this review

- Albouy, D., Graf, W., Kellogg, R. and Wolff, H., 2016. Climate amenities, climate change, and American quality of life. *Journal of the Association of Environmental and Resource Economists*, 3(1), pp.205-246.
- Bayer, P., Keohane, N. and Timmins, C., 2009. Migration and hedonic valuation: The case of air quality. *Journal of Environmental Economics and Management*, 58(1), pp.1-14.
- Fezzi, C., Harwood, A.R., Lovett, A.A. and Bateman, I., 2015. The environmental impact of climate change adaptation on land use and water quality. *Nature Climate Change*, 5(3), pp.255-260.
- Hashida, Y., and D.J. Lewis. 2019. The intersection between climate adaptation, mitigation, and natural resources: An empirical analysis of forest management. *Journal of the Association of Environmental and Resource Economists*, 6(5): 893-926.
- Hashida, Y., Withey, J., Lewis, D.J., Newman, T. and Kline, J.D., 2020. Anticipating changes in wildlife habitat induced by private forest owners’ adaptation to climate change and carbon policy. *PLOS One*, 15(4), e0230525.
- Kuminoff, N.V., Smith, V.K. and Timmins, C., 2013. The new economics of equilibrium sorting and policy evaluation using housing markets. *Journal of economic literature*, 51(4), pp.1007-62.
- Lawler, J.J., Lewis, D.J., Nelson, E., Plantinga, A.J., Polasky, S., Withey, J.C., Helmers, D.P., Martinuzzi, S., Pennington, D. and Radeloff, V.C., 2014. Projected land-use change impacts on ecosystem services in the United States. *Proceedings of the National Academy of Sciences*, 111(20), pp.7492-7497.
- Mendelsohn, R.O. and Massetti, E., 2017. The use of cross-sectional analysis to measure climate impacts on agriculture: theory and evidence. *Review of Environmental Economics and Policy*, 11(2), pp.280-298.
- Ortiz-Bobea, A., 2020. The role of nonfarm influences in Ricardian estimates of climate change impacts on US agriculture. *American Journal of Agricultural Economics*, 102(3), pp.934-959.
- Sieg, H., Smith, V.K., Banzhaf, H.S. and Walsh, R., 2004. Estimating the general equilibrium benefits of large changes in spatially delineated public goods. *International Economic Review*, 45(4), pp.1047-1077.

Reviewer #2 (Remarks to the Author):

Dear authors,

I enjoyed reading your manuscript and learn about your results. In my view, your work on disaggregating benefits of ecosystem services according to different groups of the population is a noteworthy and important contribution to the ecosystem services community but also to policy- and decision-makers in the US and beyond.

Throughout your manuscript you say that you illustrate changes between 2020 and 2100. However, if I understand right, your baseline data for the ecosystem services is never 2020 but sometime in the years before. Either clarify how you start with 2020 data or specify that you start 'nowadays' with your calculations. You could for example add in the supplementary data a table listing the different data sources and the year of the data you are using. Along these lines I would recommend to avoid the term '80 years' and use formulation like 'until 2100' and 'this century'.

As your title indicates, you (and I) find the most report-worthy information the changes across different groups of population as presented in Figure 4. Unfortunately, I didn't understand how you exactly calculated the population data and what you based it on. I assume you based it on an US census. A few sentences in the main text and a bit more in the methods would help the reader how you came up with the calculations and how they worked. I understand how you disaggregated beneficiaries (l.356-375) but I am not sure what you based it on.

Throughout the manuscript you switch between 'carbon storage' and 'climate regulation' as the ecosystem service. I would recommend using one, preferable 'climate regulation' which would then also cover your demand and benefit aspects.

You mention 'benefits nationally' (l.166,241), but you don't describe how you calculated them. Please describe on which level you calculated results (e.g. county) and which other levels are just aggregated or if everything is individually calculated.

Figure 3 (and beyond): your four quadrants indicate that the top left and bottom right are a mismatch, while the bottom left and top right are remaining in relationship. However, as you describe in your discussion, it is very likely that you start nowadays already with a mismatch and these changes might have big impacts. In addition if we take any of your four colored boxes, it matters if I have e.g. a 95% decrease in ES supply but only a 10% decrease in demand. I would welcome a more nuanced view or at least a reflection in the discussion.

In your methods you mention 'decadal changes' multiple times. Could you elaborate this fact a bit more – I assume this has to do with your models and how you reach 2100. It would be great to have some more information in the methods section about it.

I have a row of smaller comments and for this, I list them as they occur in the manuscript.

l.81/2: I am not sure why demand would increase necessarily? If population is just moving from rural to urban areas, how does this affect demand?

Table 1: You list here 'Key Beneficiaries'. I understand your goal for this table, but since you later disaggregate beneficiaries in other groups not in those four groups, I would recommend you delete the column to avoid confusion. I was waiting the whole time that you would come back to them.

Figure 2: is the bar of A2-Air quality going beyond the graph? List in the figure or at least in the caption that indicators for the benefits.

Figure 3: you could work with a, b, c, d, for easier reference in the manuscript. I wouldn't use 'most extreme' for A2, above (l.147/8) you mention it is the most realistic

l.194: what do you mean with globally constant?

l.197: do you mean carbon storage or carbon emissions?

Figure 4: Assuming this is the most important figure in your manuscript, I would recommend having boxes that show the whole bar or that let us know how long they are, include a x-axis unit, and move the legend out of the Air Quality/ Density box. Is there a way to illustrate what you show on a map to learn where the counties are that lose most? Or is there any other way to illustrate what you show in a different way? Currently Figure 3 is definitely the bigger key catcher than Figure 4.

l.226/7: is this for Black or Latinx people or combined?
l.229: I wonder if these regions are too big to find patterns.
l.316: this century not the coming, next one, no?
l.317: which stakeholders are you referring to?

Reviewer #3 (Remarks to the Author):

I have difficulty seeing the novelty of this paper. The authors use existing land cover change scenarios, and mostly appropriate but standard methods for ecosystem service mapping to get insight in future changes of ES across the United States of America. The impacts are disaggregated across several societal groups, showing that "projected losses of ecosystem services disproportionately affect non-white and lower-income populations". I think this finding provides too little novelty compared to recent papers that have done comparable analyses, such as {Chaplin-Kramer, 2019, ch008}{Ma, 2019, ma043}{Mullin, 2018, mu011}.

Then, there are several flaws or unfortunate choices regarding the methods or indicators. Methods are available to track the location of climate change impacts, and it would be straightforward to couple these to socio-economic patterns. Not disaggregating climate regulation is a missed opportunity. Furthermore, the conceptual framing of the paper (discussing demand, supply, benefits) is a bit outdated – see {Chaplin-Kramer, 2019, ch008}.

Specific comments:

L179-180: How realistic is the assumption of the constant proportions of crop types at county level? Total demand for specific crops will differ between scenarios, where most importantly the amount of soy for feed will vary. This will affect the proportions. Also, global markets will affect domestic demands for crops, which will affect the proportions within countries as well.

Figure 3 is very difficult to read and interpret. Especially the lighter hues are difficult to discern, and I don't see purple areas outlined in red. I do see many blue areas outlined in red, though.

Figure 4: legend overlaps with the air quality A2 Suburban bar.

L248-251: I don't get the point here. Many of the spatially disaggregated ES studies at pixel level work at such a detailed resolution that differences are clear at district level or sometimes more detailed. This is a level that can be easily coupled to socio economic data, and that's done increasingly. I'm afraid you're not on-top of the literature and are overselling your achievements. See e.g. {Ma, 2019, ma043}{Mullin, 2018, mu011}.

L259-261: that's not that surprising, as these counties need expansion of built-up areas and converting natural land, if low-value, is often easier than expropriating. Why do you put the solution to this problem on the shoulders of conservation organizations (L319) rather than urban planners?

*** Line numbers listed below correspond to the clean word document, not the document with track changes.**

Reviewer #1 (Remarks to the Author):

This paper examines how projected changes in U.S. ecosystem services affect different demographic groups during the period 2020 to 2100. The authors combine government projections of land-use with similar projections of population and income, and then translate the land-use change projections to a set of ecosystem services – carbon storage, air quality, crop pollination, and West-Nile virus protection. Results indicate that projected losses in ecosystem service will disproportionately affect non-white, low-income groups, especially in rural regions. This large-scale analysis of the effects of ecosystem services on different demographic groups is novel in the literature, and sheds light on an issue that is important to conservation science. I am generally positive about the manuscript, though there are multiple issues that need to be addressed.

Main comment #1: carbon storage and crop pollination services should be dropped from the analysis, which should be re-focused on the two ES that have distributional consequences and which best suits the authors' methods.

We thank the reviewer for their comments related to the carbon storage and crop pollination analyses. At the suggestion of the editor, we chose to retain carbon storage and crop pollination in the paper, but reconsidered several of our initial model assumptions. Based on these suggestions made by the reviewer, we made significant revisions to the manuscript by reframing and reanalyzing these two ES.

- Crop pollination services
 - The beneficiaries are farmers. The Census of Agriculture (2017) says 95.4% of U.S. producers are white (https://www.nass.usda.gov/Publications/Highlights/2019/2017Census_Farm_Producers.pdf). But the U.S. Census says only 76% are white.
 - The authors appear to use the population demographics to apportion the projected changes in crop pollination services (Fig. 4), but this is quite wrong since most farmers are white and it's unclear how the demographics of farmers differs from the counties where they reside.
 - A similar problem occurs with income, where it is not clear that income of farmers matches that of the population, and yet that's what appears to be assumed in Fig. 4. The USDA Economic Research Service has more information on incomes of farming households that might be useful

<https://www.ers.usda.gov/data-products/ag-and-food-statistics-charting-the-essentials/farming-and-farm-income/>).

- Given how the authors are computing the distributional aspects of changes in ES, and given that the beneficiaries of crop pollination services are farmers whose demographics differ from the broader population, I think the authors should consider dropping crop pollination services unless they can develop a more accurate way to measure demographics of this sub-group.

These are excellent points, and we appreciate the reviewer for raising them. In response, we have substantially revised the way we assess crop pollination services and their benefits. In the manuscript, we detail these modifications to our analysis in lines 422 – 432 and updated Figure 4. These changes to the analysis had little impact on our results.

To summarize, we used data from the 2012 USDA NASS Census and the US Census to conduct an additional analysis comparing farmer operator (i.e. farm owners) demographics and the demographics of the county in which their farm is located. On average, farm operators are approximately 15% more white than the general population of the county (Figure S2). To account for this difference in the distributional analysis for crop pollination, we applied county-specific scalars to adjust the population demographic projections accordingly. For example, a county that is 50% white, 30% Black, and 20% Latinx. When recalculated, those proportions change, such that the farmer operator demographics are 57.5% white, 25.5% Black, and 17% Latinx.

We also added additional text to the Methods section (lines 432 – 435), noting that, “Farmer income is also on average lower than the general population; however, because we bin counties by income quintile, we assume that the spatial distribution of farmer incomes matches that of the general population (e.g. low-income farmers operate in lower income counties, and vice-versa).”

- Carbon storage
 - I don't see why the authors include this service, since the beneficiaries include the global population. The portion of Fig. 3 for carbon storage simply presents the areas where carbon storage is projected to decline. But this has already been analyzed in past papers (e.g. the Lawler et al. 2014 study you cite), and the present manuscript doesn't really add anything new.
 - My suggestion is for the authors to drop carbon storage since it's not a part of the distributional effects of ES change (Fig. 4), which is this paper's main story.

We absolutely agree that the global benefits of carbon storage make this ES distinct in its spatial dynamics. Indeed, during the initial conceptualization of this paper, we had several discussions about the utility of including climate regulation and how best to structure that part of the analysis. These comments have prompted another discussion, and we decided to more carefully frame climate regulation as a useful contrast to the other ES.

We added the following paragraph (lines 307 – 318), to provide greater insight into our thinking on this:

“Climate regulation and crop pollination present interesting contrasts with the other ES. Although damages from climate change are expected to be highly variable across the US ¹, the magnitude and distribution of these impacts are unrelated to the locations of changes in carbon storage. As such, marginal changes in storage have the same impact on global climate change, and therefore have the same aggregate value to society (as represented by the social cost of carbon), regardless of where they occur. This is in contrast with the other three services, where benefits depend on the locations of the beneficiaries and changes in ES supply relative to each other [...].”

In addition, this comment and a comment raised by Reviewer #3 led us to reconsider some of our assumptions around the *demand* for and *benefits* of climate regulation services. Instead of measuring demand for climate regulation using the social cost of carbon, an aggregate estimate of the marginal damages from climate change globally, we revised the analysis to apply spatially-explicit (i.e. county-level) estimates of climate change damage costs. These estimates are derived from Hsiang et al. ¹ The Methods section on climate regulation demand and benefits has been completely re-written, describing these changes in detail (lines 502 – 529). We also updated Figures 2, 3, and 4, and the accompanying Results text based on these changes.

Main comment #2: The land-use and demographic projections that the authors use have major limitations that could be better discussed. I know that analyses of this scale must hold their nose and make huge assumptions (I've done the same), but I do think more discussion could be added to give the reader a better intuitive sense of the importance of these assumptions.

- Migration response to changes in ecosystem services
 - This needs more enhanced discussion, as it something ignored in your modeling.
 - The economics literature has produced much evidence that people's location decisions are related to local amenities, and air quality has received particular attention (Sieg et al. 2004; Bayer et al. 2009). Reductions in air quality in location

X relative to location Y induce migration from X to Y, and corresponding housing price reductions in X and increases in Y.

- Your current framework ignores such location adjustments. I understand why you ignore such adjustments (which would be really challenging at this scale), but I think this needs a more extended discussion. Using your language, a reduction in the “supply” of an ES in a region might induce a reduction in the “demand” of an ES by inducing out-migration or lower in-migration to the same region. How this would play out in terms of distributional effects would be a fruitful future research topic – would it exacerbate the distributional effects? E.g. past literature indicates that pollution increases induce out-migration and corresponding changes in housing prices, which will feed back and influence population and demographic projections.

We thank the reviewer for this comment. We added the following paragraph to the Discussion section to further explore the implications of migratory responses to changes in ecosystem services (lines 287 – 295):

“Emigration is another possible response to declines in the supply of ES ². For instance, variation in local and regional air quality has been found to induce household migration ³. Similarly, increasing frequency and severity of flooding has triggered voluntary buyouts of flood-prone properties ⁴. These examples raise concern that the decrease in ES supply in one locale may simply displace the demand for that ES from that location to another. In the case of government-sponsored buyout programs, decisions regarding which properties to acquire have major social justice implications ⁵. However, when migration occurs in the absence of government assistance, the financial costs to households are often overwhelming, resulting in further vulnerability to environmental hazards and the loss of ES.”

- Choice of land-use change projection and no link to climate change
 - The Sohl et al. (2016 Ecol Modeling) paper you cite notes that land-use projections vary significantly across different models and methods. I suggest adding a discussion of how the FOR-SCE projections might give different answers compared to some other candidates discussed in that article. Based on the Sohl et al. article, can you say whether other land-use projections would give more optimistic or pessimistic results in terms of disproportionate impacts? This deserves more mention than your one-sentence on p. 6 about “uncertainty”.

Good suggestion. We added the following sentences exploring this point further (lines 325 – 329):

“As compared to other land cover projection datasets based on the SRES scenarios, such as those developed by Wear ⁶, Bierwagen, et al. ⁷, and Strengers, et al. ⁸, the FORE-SCE model projections^{9,10} we use encompass greater variability across scenarios. We therefore would assume that our results include a wider range of possible outcomes than if we were to use of these other datasets.”

- I’m glad that you note that the lack of climate change impacts on the land use projections is a problem. There is, as yet, no large-scale land-use projections linked to climate change for the U.S. However, you should note that there is progress on this in the UK (Fezzi et al. 2015), and there is work that models land-use change within western U.S. forests in response to climate change adaptation by landowners (Hashida and Lewis 2019), with corresponding impacts on ecosystem services (Hashida et al. 2020). For the U.S., there is also evidence that climate change will alter urban quality of life (Albouy et al. 2016), with potential for shifting migration patterns related to my earlier comment about migration. And there has been a very active economics literature on climate impacts on farmland values (e.g. see the review by Mendelsohn and Massetti 2017; or recent paper by Ortiz-Bobea 2020). So, we know that economic returns to different land uses affect land-use change (Lawler et al. 2014), and the above literature indicates that climate change could affect land use decisions by altering these returns in a variety of directions that the FOR-SCE projections ignore. An intuitive mention of the land-use climate change link could really help the reader understand the limitations of your selection of land-use projections.

We agree, and we revised the original paragraph in the Discussion acknowledging the lack of climate change impacts to the following (lines 345 – 354):

“Capturing the impacts of climate change on land cover change, ES supply, and ES demand is another important area of future research. Neither of the land cover or population datasets we use explicitly account for climate change in their projections, and in none of our ES models do we include climatic variables. Despite these limitations, increasing evidence suggests that climate change adaptation may lead to significant changes in land use, with corresponding impacts on ES ^{11,12}. Moreover, climate change is expected to negatively affect ecological processes mediating ES supply^{13,14}, as well as shift ES demand through migratory responses of beneficiaries. Separate from the direct impacts on ES, damages from climate change are expected to

disproportionately affect poorer regions of the US, thus increasing preexisting inequality¹. Combined, these trends raise concern that our results indicating future disparities in ES are likely underestimates.”

References cited in this review

Albouy, D., Graf, W., Kellogg, R. and Wolff, H., 2016. Climate amenities, climate change, and American quality of life. *Journal of the Association of Environmental and Resource Economists*, 3(1), pp.205-246.

Bayer, P., Keohane, N. and Timmins, C., 2009. Migration and hedonic valuation: The case of air quality. *Journal of Environmental Economics and Management*, 58(1), pp.1-14.

Fezzi, C., Harwood, A.R., Lovett, A.A. and Bateman, I., 2015. The environmental impact of climate change adaptation on land use and water quality. *Nature Climate Change*, 5(3), pp.255-260.

Hashida, Y., and D.J. Lewis. 2019. The intersection between climate adaptation, mitigation, and natural resources: An empirical analysis of forest management. *Journal of the Association of Environmental and Resource Economists*, 6(5): 893-926.

Hashida, Y., Withey, J., Lewis, D.J., Newman, T. and Kline, J.D., 2020. Anticipating changes in wildlife habitat induced by private forest owners' adaptation to climate change and carbon policy. *PLOS One*, 15(4), e0230525.

Kuminoff, N.V., Smith, V.K. and Timmins, C., 2013. The new economics of equilibrium sorting and policy evaluation using housing markets. *Journal of economic literature*, 51(4), pp.1007-62.

Lawler, J.J., Lewis, D.J., Nelson, E., Plantinga, A.J., Polasky, S., Withey, J.C., Helmers, D.P., Martinuzzi, S., Pennington, D. and Radeloff, V.C., 2014. Projected land-use change impacts on ecosystem services in the United States. *Proceedings of the National Academy of Sciences*, 111(20), pp.7492-7497.

Mendelsohn, R.O. and Massetti, E., 2017. The use of cross-sectional analysis to measure climate impacts on agriculture: theory and evidence. *Review of Environmental Economics and Policy*, 11(2), pp.280-298.

Ortiz-Bobea, A., 2020. The role of nonfarm influences in Ricardian estimates of climate change impacts on US agriculture. *American Journal of Agricultural Economics*, 102(3), pp.934-959.

Sieg, H., Smith, V.K., Banzhaf, H.S. and Walsh, R., 2004. Estimating the general equilibrium benefits of large changes in spatially delineated public goods. *International Economic Review*, 45(4), pp.1047-1077.

Reviewer #2 (Remarks to the Author):

Dear authors,

I enjoyed reading your manuscript and learning about your results. In my view, your work on disaggregating benefits of ecosystem services according to different groups of the population is a noteworthy and important contribution to the ecosystem services community but also to policy- and decision-makers in the US and beyond.

Throughout your manuscript you say that you illustrate changes between 2020 and 2100. However, if I understand right, your baseline data for the ecosystem services is never 2020 but sometime in the years before. Either clarify how you start with 2020 data or specify that you start ‘nowadays’ with your calculations. You could for example add in the supplementary data a table listing the different data sources and the year of the data you are using. Along these lines I would recommend to avoid the term ‘80 years’ and use formulation like ‘until 2100’ and ‘this century’.

Thank you for this recommendation. We changed the references to ‘the next 80 years’ to ‘until 2100’ or ‘this century’ throughout the manuscript.

As your title indicates, you (and I) find the most report-worthy information the changes across different groups of population as presented in Figure 4. Unfortunately, I didn’t understand how you exactly calculated the population data and what you based it on. I assume you based it on an US census. A few sentences in the main text and a bit more in the methods would help the reader how you came up with the calculations and how they worked. I understand how you disaggregated beneficiaries (1.356-375) but I am not sure what you based it on.

We appreciate this suggestion. In response, we made revisions to better describe the land cover and population data projections (lines 106 – 116; 376 – 385):

“Our ES models are driven by existing datasets predicting future changes in land cover and population across the US between 2020 and 2100. While several land cover and population projections exist ^{15,16}, we selected datasets with common underlying assumptions and that are used by US federal government agencies. For land cover projections, we used the United States Geological Survey (USGS) FOREcasting SCENARIOS of Land-Use Change (FORE-SCE) dataset ^{9,10} (Figure 1). For population projections, we used the United States Environmental Protection Agency (US EPA) Integrated Climate and Land Use Scenarios (ICLUS) dataset ⁷ (Figure 1). The population projections were then coupled with county-level income ¹⁷ and demographic ^{16,18} projections. These projections are modulated by four alternative future scenarios (Figure 1), specified by the IPCC

Special Report on Emissions Scenarios (SRES). See Methods for more details on the assumptions and development of these scenarios.”

“Both land cover and population datasets are modulated by alternative socioeconomic and climate scenarios, representing various pathways through which social, economic, political, and environmental trends may affect land cover and population trajectories. These scenarios make varying assumptions about the degree to which the economy is globalized and regulated for environmental protection, and have implications for population growth rates, GDP growth rates, technological innovation, energy sources, and natural resource protection (Figure 1). The FORE-SCE and ICLUS datasets are based on the IPCC Special Report on Emissions Scenarios (SRES), while the income and demographic projections are based on the Shared Socioeconomic Pathways (SSPs). To ensure consistency across datasets, we map the SSPs onto the SRES based on recommendations presented by van Vuuren and Carter ¹⁹.”

Throughout the manuscript you switch between ‘carbon storage’ and ‘climate regulation’ as the ecosystem service. I would recommend using one, preferable ‘climate regulation’ which would then also cover your demand and benefit aspects.

Thank you for raising the point, as we can understand that the inconsistency in terminology may confuse readers. We revised the manuscript to refer to the ecosystem service as ‘climate regulation’, as opposed to ‘carbon storage’.

You mention ‘benefits nationally’ (l.166,241), but you don’t describe how you calculated them. Please describe on which level you calculated results (e.g. county) and which other levels are just aggregated or if everything is individually calculated.

We added a sentence in the caption for Figure 2, specifying that, “Benefits for each service were calculated at the county-level, then summed at the national-level.”

Figure 3 (and beyond): your four quadrants indicate that the top left and bottom right are a mismatch, while the bottom left and top right are remaining in relationship. However, as you describe in your discussion, it is very likely that you start nowadays already with a mismatch and these changes might have big impacts. In addition if we take any of your four colored boxes, it matters if I have e.g. a 95% decrease in ES supply but only a 10% decrease in demand. I would welcome a more nuanced view or at least a reflection in the discussion.

Yes, we agree that it is important to acknowledge this point. In the Results section, we added the following sentences further discussing ES mismatches (lines 147 – 150):

“ES mismatches are expected to occur in counties where supply decreases and demand increases between 2020 and 2100 (i.e. purple-colored counties, Figure 3; Figure S1). Among these counties, the severity of ES mismatches depends on the relative magnitude of changes in ES supply and demand.”

We also added the following to the Figure 3 caption: “This color scheme does not distinguish the severity of projected mismatches between ES supply and demand. For instance, a 90% decrease in supply and a 90% increase in demand will result in a greater mismatch than a 10% decrease in supply and a 10% increase in demand.”

In your methods you mention ‘decadal changes’ multiple times. Could you elaborate this fact a bit more – I assume this has to do with your models and how you reach 2100. It would be great to have some more information in the methods section about it.

We recognize that our use of the word ‘decadal’ may have been confusing. To address this, we removed any reference to ‘decadal’ timesteps from the methods section, as we only calculate changes between the years 2020 and 2100.

I have a row of smaller comments and for this, I list them as they occur in the manuscript.

1.81/2: I am not sure why demand would increase necessarily? If population is just moving from rural to urban areas, how does this affect demand?

We revised this sentence by clarifying that ES mismatches are expected to occur in counties where supply decreases and demand increases (lines 83-85).

Table 1: You list here ‘Key Beneficiaries’. I understand your goal for this table, but since you later disaggregate beneficiaries in other groups not in those four groups, I would recommend you delete the column to avoid confusion. I was waiting the whole time that you would come back to them.

We thank the reviewer for this comment, and now realize that after introducing the “Key Beneficiaries” in Table 1 we seldom reference them in other parts of the manuscript. Rather than delete the column from the table, we provide additional discussion of these beneficiaries at several points throughout the manuscript.

Figure 2: is the bar of A2-Air quality going beyond the graph? List in the figure or at least in the caption that indicators for the benefits.

We adjusted the y-axis limits on Figure 2 so that the A2-Air Quality bar is no longer extending beyond the limits of the plot. We also modified the caption for Figure 2 to state that, “The indicators used for each of the benefits are described in Table 1.”

Figure 3: you could work with a, b, c, d, for easier reference in the manuscript. I wouldn't use 'most extreme' for A2, above (1.147/8) you mention it is the most realistic

Thank you for this suggestion. To make the maps easier to reference, we added A, B, C, and D labels. We also revised the figure caption the state that scenario A2 is the “most realistic”, rather than the “most extreme.”

l.194: what do you mean with globally constant?

We clarified the wording in the sentence to indicate that:

“[...] marginal changes in storage have the same impact on global climate change, and therefore have the same aggregate value to society (as represented by the social cost of carbon), regardless of where they occur” (lines 310 – 312).

l.197: do you mean carbon storage or carbon emissions?

Clarified that we are referring to carbon storage.

Figure 4: Assuming this is the most important figure in your manuscript, I would recommend having boxes that show the whole bar or that let us know how long they are, include a x-axis unit, and move the legend out of the Air Quality/ Density box. Is there a way to illustrate what you show on a map to learn where the counties are that lose most? Or is there any other way to illustrate what you show in a different way? Currently Figure 3 is definitely the bigger eye catcher than Figure 4.

Thank you for these recommendations. We made the following changes to Figure 4:

- **Added x-axis labels, specifying that the unit is percent change between 2020 and 2100.**
- **Moved the legend so that is no longer overlapping with the bar in Air Quality / Density.**
- **Change the label for the third row of plots from “Race” to “Race / Ethnicity”.**
- **Added the following sentence to the figure caption, “The lower and upper limits of the x-axis for each subplot are set to -150 and 150 to facilitate comparison across subplots; however, some bars extend beyond these bounds.”**

1.226/7: is this for Black or Latinx people or combined?

Clarified in line 225 that we are referring to non-white people.

1.229: I wonder if these regions are too big to find patterns.

It seems possible. We added an acknowledgment that these are in fact very broad geographic regions.

1.316: this century not the coming, next one, no?

Correct. Changed phrasing from “coming century” to “this century”.

1.317: which stakeholders are you referring to?

Clarified in line 357 that we are referring specifically to land owners.

Reviewer #3 (Remarks to the Author):

I have difficulty seeing the novelty of this paper. The authors use existing land cover change scenarios, and mostly appropriate but standard methods for ecosystem service mapping to get insight in future changes of ES across the United States of America. The impacts are disaggregated across several societal groups, showing that "projected losses of ecosystem services disproportionately affect non-white and lower-income populations". I think this finding provides too little novelty compared to recent papers that have done comparable analyses, such as Chaplin-Kramer et al. (2019), Ma et al. (2019), and Mullin et al. (2018).

Thank you for raising this concern. To address this, we added the following paragraph highlighting the novelty and importance of this work, and discussing the ways in which it diverges and builds upon previous studies (lines 123 – 133):

“This study is the first to project changes in multiple ES benefits at the national scale and assess the distribution of those benefits among demographic and socioeconomic groups. Calls have been made for better disaggregation of ES benefits among beneficiary groups²⁰⁻²², yet to date, relatively few studies have done so²³. Notably, Chaplin-Kramer et al.²⁴ project changes in ES globally, but do not distinguish projected impacts among demographic or socioeconomic groups. Additionally, Ma et al.²⁵ and Mullin et al.²⁶ highlight that socially vulnerable groups have less access to natural capital and are at greater risk of losing ES benefits in the future. Our analysis builds on these prior studies by disaggregating ES benefits according to specific racial and ethnic groups, as well as income quantiles. This is particularly important in the US, where historical political, economic, and social trends have perpetuated and reinforced inequality specifically along lines of race, ethnicity, and class.”

Then, there are several flaws or unfortunate choices regarding the methods or indicators. Methods are available to track the location of climate change impacts, and it would be straightforward to couple these to socio-economic patterns. Not disaggregating climate regulation is a missed opportunity.

Thank you for this comment. Please see our response to Reviewer #1’s first main comment related to carbon storage.

Furthermore, the conceptual framing of the paper (discussing demand, supply, benefits) is a bit outdated – see Chaplin-Kramer et al. (2019).

During the original conceptualization of this analysis, we considered the merits and limitations of several previously developed conceptual frameworks, including those by

Chaplin-Kramer et al. (2019)²⁴ and Ma et al. (2019)²⁵, as well as other recent studies. While the conceptual framework developed by Chaplin-Kramer et al. (2019)²⁴ is noteworthy, the variables that they actually quantify are very similar to what we quantify in this analysis and what is quantified by Ma et al. (2019)²⁵. For example, in Figure 2 in Chaplin-Kramer et al. (2019)²⁴, they show maps corresponding to 1) “Nature’s Contributions” (i.e. supply), 2) “Populations Exposed” (i.e. demand), and 3) “Nature’s Contributions to People” (i.e. benefit). We acknowledge the contributions made by Chaplin-Kramer et al. (2019)²⁴ at several points throughout the manuscript, however, we believe that the conceptual framework we apply is simpler, easier to understand, and better suited to the purposes of our analysis.

Specific comments:

L179-180: How realistic is the assumption of the constant proportions of crop types at county level? Total demand for specific crops will differ between scenarios, where most importantly the amount of soy for feed will vary. This will affect the proportions. Also, global markets will affect domestic demands for crops, which will affect the proportions within countries as well.

This is a good question. We added a caveat in the Methods acknowledging that this assumption is probably not very realistic and that it is a limitation of our analysis (lines 561 – 563):

“While demand for specific crops is likely to change in future years, modeling how the proportion of crop types are expected to change through time is beyond the scope of our analysis. To estimate the benefit of pollination services, we multiplied the supply and demand metrics.”

Figure 3 is very difficult to read and interpret. Especially the lighter hues are difficult to discern, and I don’t see purple areas outlined in red. I do see many blue areas outlined in red, though.

We have gone through many iterations of color palettes for this figure, but realize that the one we chose may not work for everyone. In an effort to makes the colors easier to discern, we adjusted the RGB values for the lighter hues, attempting to make them more distinctive. We also added more red to the color that we called “purple” and the reviewer called “blue”, with the hope of making it appear more “purple” and less “blue”.

Figure 4: legend overlaps with the air quality A2 Suburban bar.

Modified Figure 4 so that the legend no longer overlaps with the air quality A2 Suburban bar.

L248-251: I don't get the point here. Many of the spatially disaggregated ES studies at pixel level work at such a detailed resolution that differences are clear at district level or sometimes more detailed. This is a level that can be easily coupled to socio economic data, and that's done increasingly. I'm afraid you're not on-top of the literature and are overselling your achievements. See e.g. Ma et al. (2019), Mullin et al. (2018).

A recent review by Mandle et al. (2020)²³ found that in most ES studies to date, ES benefits are either disaggregated spatially (e.g. pixels across a landscape), regionally, or not at all, and are rarely disaggregated among beneficiary groups (lines 251 – 252). While we agree that ES data at the pixel-level could easily be coupled to socioeconomic data, this is relatively uncommon, and to our knowledge, has not been done before at the scale of the United States. However, as the reviewer mentions, several recent studies have made advances in coupling ES benefits with socioeconomic data. As such, we added the following sentence (lines 260 – 264), acknowledging that, “[...] several recent studies have demonstrated the potential for novel tools and techniques, such as open-access social media datasets (e.g. Hamstead et al. ²⁷), agent-based modeling (e.g. Miyasaka et al. ²⁸), and social vulnerability indices (e.g. Mullin et al. ²⁶), in evaluating the distribution of ES benefits among beneficiary groups ²².”

L259-261: that's not that surprising, as these counties need expansion of built-up areas and converting natural land, if low-value, is often easier than expropriating. Why do you put the solution to this problem on the shoulders of conservation organizations (L319) rather than urban planners?

Fair point. We added that urban planners also have an important role to play in addressing these issues (line 360).

References (in the order that they appear in the responses to the reviewers' comments)

- 1 Hsiang, S. *et al.* Estimating economic damage from climate change in the United States. *Science* **356**, 1362-1369 (2017).
- 2 Renaud, F. G., Dun, O., Warner, K. & Bogardi, J. A decision framework for environmentally induced migration. *International Migration* **49**, e5-e29 (2011).
- 3 Bayer, P., Keohane, N. & Timmins, C. Migration and hedonic valuation: The case of air quality. *J. Environ. Econ. Manage.* **58**, 1-14 (2009).
- 4 Mach, K. J. *et al.* Managed retreat through voluntary buyouts of flood-prone properties. *Science Advances* **5**, eaax8995 (2019).
- 5 Siders, A. R. Social justice implications of US managed retreat buyout programs. *Climatic Change* **152**, 239-257 (2019).
- 6 Wear, D. N. Forecasts of county-level land uses under three future scenarios: a technical document supporting the Forest Service 2010 RPA Assessment. *Gen. Tech. Rep. SRS-141. Asheville, NC: US Department of Agriculture Forest Service, Southern Research Station. 41 p.* **141**, 1-41 (2011).
- 7 Bierwagen, B. G. *et al.* National housing and impervious surface scenarios for integrated climate impact assessments. *Proceedings of the National Academy of Sciences* **107**, 20887-20892 (2010).
- 8 Strengers, B., Leemans, R., Eickhout, B., de Vries, B. & Bouwman, L. The land-use projections and resulting emissions in the IPCC SRES scenarios scenarios as simulated by the IMAGE 2.2 model. *GeoJournal* **61**, 381-393 (2004).
- 9 Sleeter, B. M. *et al.* Scenarios of land use and land cover change in the conterminous United States: Utilizing the special report on emission scenarios at ecoregional scales. *Global Environmental Change* **22**, 896-914 (2012).
- 10 Sohl, T. L. *et al.* Spatially explicit modeling of 1992–2100 land cover and forest stand age for the conterminous United States. *Ecol. Appl.* **24**, 1015-1036 (2014).
- 11 Fezzi, C., Harwood, A. R., Lovett, A. A. & Bateman, I. J. in *Building a Climate Resilient Economy and Society* (Edward Elgar Publishing, 2017).
- 12 Hashida, Y. & Lewis, D. J. The intersection between climate adaptation, mitigation, and natural resources: An empirical analysis of forest management. *Journal of the Association of Environmental and Resource Economists* **6**, 893-926 (2019).
- 13 Mooney, H. *et al.* Biodiversity, climate change, and ecosystem services. *Current Opinion in Environmental Sustainability* **1**, 46-54 (2009).
- 14 Runting, R. K. *et al.* Incorporating climate change into ecosystem service assessments and decisions: a review. *Glob. Chang. Biol.* **23**, 28-41 (2017).
- 15 Sohl, T. L., Wimberly, M. C., Radeloff, V. C., Theobald, D. M. & Sleeter, B. M. Divergent projections of future land use in the United States arising from different models and scenarios. *Ecol. Modell.* **337**, 281-297 (2016).
- 16 Hauer, M. E. Population projections for US counties by age, sex, and race controlled to shared socioeconomic pathway. *Scientific Data* **6**, 190005 (2019).
- 17 Wear, D. N. & Prestemon, J. P. Spatiotemporal downscaling of global population and income scenarios for the United States. *PLoS One* **14** (2019).
- 18 NASS, U. Census of Agriculture. *US Department of Agriculture, National Agricultural Statistics Service, Washington, DC* (2012).

- 19 van Vuuren, D. P. & Carter, T. R. Climate and socio-economic scenarios for climate change research and assessment: reconciling the new with the old. *Climatic Change* **122**, 415-429 (2014).
- 20 Bennett, E. M. *et al.* Linking biodiversity, ecosystem services, and human well-being: three challenges for designing research for sustainability. *Current Opinion in Environmental Sustainability* **14**, 76-85 (2015).
- 21 Daw, T., Brown, K., Rosendo, S. & Pomeroy, R. Applying the ecosystem services concept to poverty alleviation: the need to disaggregate human well-being. *Environ. Conserv.* **38**, 370-379 (2011).
- 22 Rieb, J. T. *et al.* When, Where, and How Nature Matters for Ecosystem Services: Challenges for the Next Generation of Ecosystem Service Models. *Bioscience* **67**, 820-833 (2017).
- 23 Mandle, L. *et al.* Increasing decision relevance of ecosystem service science. *Nature Sustainability*, doi:10.1038/s41893-020-00625-y (2020).
- 24 Chaplin-Kramer, R. *et al.* Global modeling of nature's contributions to people. *Science* **366**, 255-258 (2019).
- 25 Ma, S., Smailes, M., Zheng, H. & Robinson, B. E. Who is vulnerable to ecosystem service change? Reconciling locally disaggregated ecosystem service supply and demand. *Ecol. Econ.* **157**, 312-320 (2019).
- 26 Mullin, K., Mitchell, G., Nawaz, N. R. & Waters, R. D. Natural capital and the poor in England: Towards an environmental justice analysis of ecosystem services in a high income country. *Landscape and Urban Planning* **176**, 10-21 (2018).
- 27 Hamstead, Z. A. *et al.* Geolocated social media as a rapid indicator of park visitation and equitable park access. *Comput. Environ. Urban Syst.* **72**, 38-50 (2018).
- 28 Miyasaka, T., Le, Q. B., Okuro, T., Zhao, X. & Takeuchi, K. Agent-based modeling of complex social-ecological feedback loops to assess multi-dimensional trade-offs in dryland ecosystem services. *Landsc. Ecol.* **32**, 707-727 (2017).

Reviewer comments, second round –

Reviewer #1 (Remarks to the Author):

I was positive about the first version of this manuscript, and I remain positive. I am satisfied with the authors' revisions in terms of how they re-computed crop pollination services and their re-written discussion that brings out some of the limitations of their analysis in a more refined way. My initial suggestion was for the authors to drop carbon storage given its global scale, but they opted for a different route. For carbon storage, I appreciate the authors' new approach of estimating the spatial pattern of damages that derive from land-use impacts on carbon storage using Hsiang et al.'s recent county-level climate change damage estimates. However, I do not understand the new metric that they use and the scale of some of the results.

- Table 1 calls the benefit measure for carbon storage as "mitigation of climate change damages, weighted by population (index)". In the methods section, the authors describe calculation of this benefit metric the following way: "To calculate this metric, we divided the aggregate supply of climate regulation (i.e. total carbon stored nationally) by county-level demand multiplied by negative one."
- So the numerator in this index is physical carbon storage ($\text{Mg C} \times \text{ha}^{-1}$), and the denominator in this index is Hsiang et al.'s GDP loss from climate change? Maybe I'm missing something, but I don't get this at all, and I suspect other readers will be confused. How does a ratio of physical carbon storage to climate-induced GDP loss translate to a benefit metric? I get what Hsiang et al. did, and I get your physical carbon storage number, but I don't get the ratio of these two things. This needs clarifying work.
- Related to this climate regulation index, how do we interpret the magnitude of the results? The lost climate regulation services on non-white individuals is over 100%. What does this mean? Since the index is hard to understand, then the magnitude of over 100% loss is harder to understand. What is a 150% loss in this metric?
- I appreciate the authors' attempt to make the paper richer with the spatial climate regulation analysis, but the new approach needs to be greatly clarified. How does a % change in land-use lead to >100% change in this benefit index? What is the baseline that is being used here? Is it 2020 levels, or something else from Hsiang et al.'s GDP calculations?

Reviewer #1

I was positive about the first version of this manuscript, and I remain positive. I am satisfied with the authors' revisions in terms of how they re-computed crop pollination services and their re-written discussion that brings out some of the limitations of their analysis in a more refined way. My initial suggestion was for the authors to drop carbon storage given its global scale, but they opted for a different route. For carbon storage, I appreciate the authors' new approach of estimating the spatial pattern of damages that derive from land-use impacts on carbon storage using Hsiang et al.'s recent county-level climate change damage estimates. However, I do not understand the new metric that they use and the scale of some of the results.

Table 1 calls the benefit measure for carbon storage as "mitigation of climate change damages, weighted by population (index)". In the methods section, the authors describe calculation of this benefit metric the following way: "To calculate this metric, we divided the aggregate supply of climate regulation (i.e. total carbon stored nationally) by county-level demand multiplied by negative one."

So the numerator in this index is physical carbon storage ($\text{Mg C} \times \text{ha}^{-1}$), and the denominator in this index is Hsiang et al.'s GDP loss from climate change? Maybe I'm missing something, but I don't get this at all, and I suspect other readers will be confused. How does a ratio of physical carbon storage to climate-induced GDP loss translate to a benefit metric? I get what Hsiang et al. did, and I get your physical carbon storage number, but I don't get the ratio of these two things. This needs clarifying work.

Related to this climate regulation index, how do we interpret the magnitude of the results? The lost climate regulation services on non-white individuals is over 100%. What does this mean? Since the index is hard to understand, then the magnitude of over 100% loss is harder to understand. What is a 150% loss in this metric?

I appreciate the authors' attempt to make the paper richer with the spatial climate regulation analysis, but the new approach needs to be greatly clarified. How does a % change in land-use lead to >100% change in this benefit index? What is the baseline that is being used here? Is it 2020 levels, or something else from Hsiang et al.'s GDP calculations?

Response to Reviewer #1

We are glad to hear that our modification of the crop pollination analysis and revision of the discussion section are satisfactory. We also appreciate that the reviewer finds merit in our decision to take an alternative approach in assessing the benefits of climate regulation.

As the reviewer points out, Hsiang et al. does not establish a direct relationship between carbon emissions and county-level climate-induced GDP loss. As a result, we are reliant on simple unitless indices to describe aspects of climate regulation demand and benefits. The questions raised by the reviewer regarding these indices are valid and reveal blind spots in our initial description of our methods.

Upon further consideration, we realized that our construction of these indices was indeed unclear and perhaps overly complicated. To address these issues, we provide additional details on how we estimate the *demand* metric (lines 506 - 520) and revised our calculation of the *benefit* metric (lines 522 - 538). In both sections, we provide examples of how the construction of these indices may affect our results.

By modifying the construction of the *benefit* metric, our results in Figure 2 and Figure 4 changed slightly, however, the overall findings of the paper remain the same. These changes are reflected throughout the Results section.

Reviewer comments, third round –

Reviewer #1 (Remarks to the Author):

I remain positive about the overall goals and message of this paper, but I go back to my original comment that the authors should drop the climate regulation analysis. After initially suggesting the authors drop climate regulation, the authors came back with a completely new climate regulation approach that tried to use Hsiang et al.'s empirical damage function. After I questioned how they used Hsiang's damage function, the authors came back with a different approach to using the damage function, which then led to different estimates. The problem is that it is exceptionally difficult to translate land-use change scenarios to changes in those damage functions, which is why the authors are having trouble with it. Here's my issue with the current approach:

- Hsiang's composite damage function aggregates multiple empirically estimated damage functions from different sectors, such as the effects of climate on crop yields and mortality. Each of these individual damage functions are highly non-linear functions of climate (e.g. Fig. 1 in that paper).
- The land-use scenarios in this paper affect carbon storage, which then affects climate, which then affects Hsiang's damaged function through non-linear effects on crop yields and mortality and the other sectors. Ideally, your carbon storage projections should first be translated into climate changes (temp, precip), and then those climate changes could be translated into the individual damage functions. Thus, using your notation, the county level damage estimates (d) are a function of carbon storage impacts on climate.
- In contrast, the approach in this paper simply multiplies the percentage change in carbon storage by Hsiang's population weighted damage function. These results could be highly misleading given the non-linearities in that damage function.

I continue to like the paper without the climate regulation material, and I think it's an excellent contribution without that material. Including this highly ad-hoc and potentially misleading climate regulation calculation really takes away from the work and diminishes the paper.

Reviewer #2 (Remarks to the Author):

Dear authors,

As in my review in November 2020, I am still of the belief that this is an important manuscript. It is well written and easy to follow which is not easy for such a complicated topic.

However, I still have some concerns about your demographic data use and analysis. I do not bring this up to annoy you or slow down your publication process. In case you want to spread your findings more widely, including involving media, I think the findings on who will lose (and benefit) are probably the most interesting for many people. As such, I think it is absolutely crucial to be clear and precise about the demographic data use and analysis. I hope that in my following comments I can clearly express my concerns and show pathways forward. I doubt that this is a major revision, rather a few sentences or minor corrections in your writing.

I am still not 100% sure where you get your data for the race category. I think it is Hauer (2019) (reference 29), but it would be fantastic if you could spell this clearly out in your method section after the first sentence l.408/9. If it is Hauer (2019) (or any other source), I would appreciate if you can explain how you come to your categories (Black, Latinx, people of color, and white compared to e.g., Hauer's White NH, Black NH, Other NH, Hispanic. You write that your POC group includes Black and Latinx people, while Hauer seems to clearly separate the four used groups. If your POC group includes Black and Latinx people, how do you ensure to avoid double counting or is it non-relevant for you?

Based on my last comment on the POC category: You use 'People of Color' as a category in your

analysis and you use the term as well in your main text. Please clarify the use in the main text – this may solve itself depending how you define the POC category.

Figure 4 – third row. It looks like Latinx, People of Color and White are the same. Comparing it with the Figure 4 in the original submission of fall 2020 I think there must be a mistake in your code?!?

In l.244 you state that rural communities are at greater risk of losing benefits from ES, which I cannot see in Fig. 4 where it looks much more to me that urban communities would lose a lot of air quality and disease control. You repeat this claim in your abstract (l.13). Please explain what I overlook or revise this statement.

REVIEWER COMMENTS

Reviewer #1 (Remarks to the Author):

I remain positive about the overall goals and message of this paper, but I go back to my original comment that the authors should drop the climate regulation analysis. After initially suggesting the authors drop climate regulation, the authors came back with a completely new climate regulation approach that tried to use Hsiang et al.'s empirical damage function. After I questioned how they used Hsiang's damage function, the authors came back with a different approach to using the damage function, which then led to different estimates. The problem is that it is exceptionally difficult to translate land-use change scenarios to changes in those damage functions, which is why the authors are having trouble with it. Here's my issue with the current approach:

- Hsiang's composite damage function aggregates multiple empirically estimated damage functions from different sectors, such as the effects of climate on crop yields and mortality. Each of these individual damage functions are highly non-linear functions of climate (e.g. Fig. 1 in that paper).
- The land-use scenarios in this paper affect carbon storage, which then affects climate, which then affects Hsiang's damaged function through non-linear effects on crop yields and mortality and the other sectors. Ideally, your carbon storage projections should first be translated into climate changes (temp, precip), and then those climate changes could be translated into the individual damage functions. Thus, using your notation, the county level damage estimates (d) are a function of carbon storage impacts on climate.
- In contrast, the approach in this paper simply multiplies the percentage change in carbon storage by Hsiang's population weighted damage function. These results could be highly misleading given the non-linearities in that damage function.

I continue to like the paper without the climate regulation material, and I think it's an excellent contribution without that material. Including this highly ad-hoc and potentially misleading climate regulation calculation really takes away from the work and diminishes the paper.

We appreciate the reviewer's thoughtfulness in working with us to improve the paper. The climate regulation analysis has now been completely removed from the paper.

Reviewer #2 (Remarks to the Author):

Dear authors,

As in my review in November 2020, I am still of the belief that this is an important manuscript. It is well written and easy to follow which is not easy for such a complicated topic.

However, I still have some concerns about your demographic data use and analysis. I do not bring this up to annoy you or slow down your publication process. In case you want to spread your findings more widely, including involving media, I think the findings on who will lose (and benefit) are probably the most interesting for many people. As such, I think it is absolutely crucial to be clear and precise about the demographic data use and analysis. I hope that in my following comments I can clearly express my concerns and show pathways forward. I doubt that this is a major revision, rather a few sentences or minor corrections in your writing.

I am still not 100% sure where you get your data for the race category. I think it is Hauer (2019) (reference 29), but it would be fantastic if you could spell this clearly out in your method section after the first sentence 1.408/9. If it is Hauer (2019) (or any other source), I would appreciate if you can explain how you come to your categories (Black, Latinx, people of color, and white compared to e.g., Hauer's White NH, Black NH, Other NH, Hispanic. You write that your POC group includes Black and Latinx people, while Hauer seems to clearly separate the four used groups. If your POC group includes Black and Latinx people, how do you ensure to avoid double counting or is it non-relevant for you?

We appreciate the reviewer for raising this concern. Indeed, clear and precise language is particularly important when discussing issues pertaining to race and ethnicity. It is our hope that the findings of this paper are shared widely, and in doing so, we made a more concerted effort to be transparent about our use and analysis of demographic data.

In response to these comments, we rewrote the paragraph in the Methods discussing the demographic groups (lines 386-391). That paragraph now reads as follows:

“Projections of demographic change are drawn from the dataset developed by Hauer (2019)²⁹. Following the classifications used this dataset, we define demographic groups based on the population of each county that is Black (non-Hispanic), Hispanic, Other (non-Hispanic), and White (non-Hispanic). Hauer²⁹ states that the Other (non-Hispanic) group refers specifically to “American Indian/Alaska Native and Asian/Pacific Islander” populations. Moreover, each group is mutually-exclusive of the others, such that there is no overlap between groups.”

We also added the following paragraph (lines 399-406):

“We acknowledge here that grouping people based on race is imperfect, as there is no biological basis for defining differences by race. In our results (see Figure 4), however, there are distinct trends differentiating predicted outcomes for the White group, versus outcomes for the Black, Hispanic, and Other groups. For summary purposes, we refer to Black, Hispanic, and Other people under a single umbrella term: “non-white”. While the term “non-white” may not be appropriate in many circumstances, we find it to be useful here specifically because the predicted outcomes for the White group are directionally opposite from the outcomes for the Black, Hispanic, and Other groups.”

Based on my last comment on the POC category: You use ‘People of Color’ as a category in your analysis and you use the term as well in your main text. Please clarify the use in the main text – this may solve itself depending how you define the POC category.

We no longer use the term the term “People of Color” and instead use the term “Non-White”. See response to above comment for more details regarding this choice.

Figure 4 – third row. It looks like Latinx, People of Color and White are the same. Comparing it with the Figure 4 in the original submission of fall 2020 I think there must be a mistake in your code?!?

There was in fact a bug in our code and we are grateful to the reviewer for identifying this error. These results have now been corrected in Figure 4 and in the text.

In l.244 you state that rural communities are at greater risk of losing benefits from ES, which I cannot see in Fig. 4 where it looks much more to me that urban communities would lose a lot of air quality and disease control. You repeat this claim in your abstract (l.13). Please explain what I overlook or revise this statement.

We thank the reviewer for pointing out this inconsistency, and have revised these statements to indicate it is *urban* communities that are at greater risk (lines 11, 229).

Reviewer comments, fourth round –

Reviewer #2 (Remarks to the Author):

Dear authors,

thank you for your revisions. I think now the issues of the demographic data have been clarified to a satisfying level.

Please note that in l. 387 there seems to be a word (maybe 'in') missing - this can be fixed in the printing proofs.

Congratulations on this manuscript!

REVIEWERS' COMMENTS

Reviewer #2 (Remarks to the Author):

Dear authors,

thank you for your revisions. I think now the issues of the demographic data have been clarified to a satisfying level.

Please note that in l. 387 there seems to be a word (maybe 'in') missing - this can be fixed in the printing proofs.

Congratulations on this manuscript!

We appreciate the reviewer for identifying this typo and have fixed it.